# mHealth Technology as a Help Tool during Breast Cancer Treatment: A Content Focus Group

**DOI:** 10.3390/ijerph20054584

**Published:** 2023-03-04

**Authors:** Angeles Fuentes, Clara Amat, Raimundo Lozano-Rubí, Santiago Frid, Montserrat Muñoz, Joan Escarrabill, Imma Grau-Corral

**Affiliations:** 1Fundación iSYS, 08028 Barcelona, Spain; 2Fundació Clínic per a la Recerca Biomèdica, 08036 Barcelona, Spain; 3Medical Informatics Unit, Hospital Clínic de Barcelona, 08036 Barcelona, Spain; 4Oncology Service, Hospital Clínic de Barcelona, 08036 Barcelona, Spain; 5Patient Xperience, Hospital Clínic de Barcelona, 08036 Barcelona, Spain; 6Hospital Clínic de Barcelona, 08036 Barcelona, Spain

**Keywords:** breast cancer, mHealth, focus group

## Abstract

Purpose: To assess the usability and preferences of the contents of mHealth software developed for breast cancer patients as a tool to obtain patient-reported outcomes (PROMs), improve the patient’s knowledge about the disease and its side effects, increase adherence to treatment, and facilitate communication with the doctor. Intervention: an mHealth tool called the Xemio app provides side effect tracking, social calendars, and a personalized and trusted disease information platform to deliver evidence-based advice and education for breast cancer patients. Method: A qualitative research study using semi-structured focus groups was conducted and evaluated. This involved a group interview and a cognitive walking test using Android devices, with the participation of breast cancer survivors. Results: The ability to track side effects and the availability of reliable content were the main benefits of using the application. The ease of use and the method of interaction were the primary concerns; however, all participants agreed that the application would be beneficial to users. Finally, participants expressed their expectations of being informed by their healthcare providers about the launch of the Xemio app. Conclusion: Participants perceived the need for reliable health information and its benefits through an mHealth app. Therefore, applications for breast cancer patients must be designed with accessibility as a key consideration.

## 1. Introduction

Breast cancer is the most common form of cancer among women [1,2]. In 2018, breast cancer mortality trends decreased by 41% due to various factors, including early diagnosis, advancements in treatment, lifestyle changes, improved nutrition, and research [3,4]. This increased survivorship highlights the importance of focusing on the long-term goals and consequences of treatment to enhance quality of life and overall well-being, promoting a proactive approach to health [5].

Technological developments in recent years have been essential in supporting methodologies for diagnosing health and cancer. These developments include the standardization of portable and wearable devices for data collection and health biomarkers, as well as advances in data analysis through artificial intelligence [6,7,8,9,10].

The use of apps to promote health and well-being has grown exponentially [11]. Smartphones facilitate the creation and development of millions of apps, including communication apps, geolocation with maps, video games, video streaming, and health apps. These can be easily downloaded from app stores and offer low-cost solutions that can be accessed by a large global population.

Using mHealth (mobile Health), also known as health apps, to support breast cancer patients during treatment and post-treatment can be a helpful complement to the usual treatment for patients [12]. mHealth can be an effective tool for obtaining patient-reported outcome measures (PROMs) that reflect patients’ perceptions of their own health. Within the framework of Value-Based Healthcare (VBHC), there is a value change from volume-driven to value-driven care, empowering patients by allowing them to report on their disease-related side effects and quality of life, and reinforcing treatment adherence [13,14,15,16,17,18,19].

PROMs are thought to be central to the understanding of the effectiveness of treatments in cancer [20], improving communication between patients and providers, patient satisfaction [21], daily life [22], and survival [23]. According to Osborn et al., a small number of mHealth applications have been used in clinical studies examining a variety of cancer types and age groups. The studies found that the positive impact was largely limited to improved symptom control, although some studies reported increased symptoms. Data on other outcomes, including health economic measures, were limited [17]. 

Xemio (www.xemio.org (accessed on 3 March 2023)) is a digital platform that comprises a website, social network, and app, providing access to a virtual environment for meetings, debates, support, and accompaniment. It was developed by Fundación ISYS, with patients as the primary focus, specifically for those with breast cancer. The project has created the Xemio app (Figure 1), an app designed by patients and doctors, and all its content is reviewed and updated by oncology professionals to help patients with their disease self-management and social issues. The platform, built for smartphones, helps patients and their families track side effects and treatments, as well as participate in activities and social events organized by various associations. The Xemio platform is endorsed by the SOLTI scientific societies dedicated to breast cancer research and the Catalan Society of Family and Community Medicine (CAMFiC). It has received support from the “la Caixa” Foundation, a Europe Horizon 2020 grant, and crowdfunding.

In order to assess the patients’ preferences for the prototype of the Xemio app, the research group decided to conduct a focus group as the first step in a series of participatory user-centered activities to develop a mobile app that is well received by patients.

The results presented in this article from the focus group are part of a larger research project. The Xemio app is integrated with the Electronic Medical Record of Hospital Clínic de Barcelona [24]. This will allow oncologists to access and interact with data recorded by patients participating in the pilot study. This integration is part of the European project “Artificial Intelligence Supporting Cancer Patients across Europe” (ASCAPE) (ClinicalTrials.gov Identifier: NCT04879563). ASCAPE aims to identify quality-of-life problems based on PROMs and support treatment recommendations.

In order to establish the design process of the Xemio app, a qualitative observational study design was previously conducted [25,26]. The study design incorporated semi-structured interviews with five patients from a local patient association [27], with the aim of identifying the desired content and features of a mobile app to assist individuals living with breast cancer [28]. The smartphone app prototype was developed with the help of an oncologist and two general practitioners belonging to the research group, and it is based on this prototype that this study was carried out.

## 2. Objectives

The aim of this focus group was to gain an in-depth understanding of the needs of breast cancer patients during treatment and assess the feasibility of a smartphone application prototype developed by a research team of patients and oncology professionals.

## 3. Methodology

Qualitative research methods provide a deeper understanding of social issues. These techniques offer more opportunities for gaining in-depth knowledge about a specific topic compared to quantitative research methods [29,30]. A focus group is a commonly used qualitative research technique [25,26] that does not require extensive resources and enables interactive feedback and suggestions from participants during the sessions. It helps to identify key areas for improvement in a product or service [31]. This study followed the flowchart of steps for conducting a focus group discussion [29].

A focus group session began with the presentation of the content to be discussed after a brief presentation by the moderator. The moderator asked the participants about their experiences. 

### 3.1. Patient Identification and Patient Recruitment

Before patient selection, it was decided that the group should be composed of breast cancer survivor patients treated at the Hospital Clinic. None of the patients were on active cancer treatment when the focus group took place. The patients invited to participate were women who represented the prototype patient cases designed for the focus group. These patients represent different age groups, between 50 and 65 years old and over 65 years old, considering that the average age of breast cancer patients is 63 in white women, combining situations of employment or unemployment and living alone or with family. Although there is a generational gap in the use of new technologies, researchers decided to model the focus group, including older adult patients, as a very suitable methodology for marginalized groups [32,33].

Due to the COVID-19 emergency in June 2020, it was challenging to recruit and invite patients to increase participation in the focus group. In accordance with the COVID-19 regulations regarding the gathering of people in enclosed spaces and hospitals, the focus group consisted of 5 participants aged between 52 and 71 years old. The archetypes of the type of patients that were of interest were the following:Patient 50–65 years old, employed;Patient 50–65 years old, unemployed;Patient > 65 years old that lives alone;Patient > 65 years old that lives with family members.

Participant identification was followed by participant recruitment. For the recruitment of participants, the collaborating oncologist drew up a list of possible candidates following the mentioned archetypes as best as possible. The oncologist contacted the candidates over the phone.

### 3.2. Hosted Focus Group Events

#### 3.2.1. Data Collection on Patient Information Needs, Services, and Activities (Session I) 

The focus group was divided into two sessions. Both sessions occurred on the same day. This first session was done without giving the patients any prior proposals of what they would evaluate in the second session, and it aimed at exploring the immediate impressions that patients have regarding information, services, and activities that they considered helpful during cancer treatment. 

During the first part of the focus group, the moderator, an experienced doctor in charge of the Patient Experience department of Hospital Clínic de Barcelona, proposed the topics to be discussed with the participants. The topics of discussion were agreed upon beforehand with two other experts that also acted as observers: an oncologist and an Information Society expert. The topics to be discussed were as follows:Treatment of symptoms;Advice on how to cope with side effects;Services or activities needed throughout the cancer process;News that would be of interest during this period.

The proposed contents of the conversation were elaborated through a thematic study group and collective alignments, considering the participants’ previous experiences. It was planned as a 50 min session.

Data collection in the first session was done by recording an audio tape (and a subsequent transcription), taking notes, and participant observation. 

#### 3.2.2. Cognitive Walkthrough Test with Users (Session II)

Cognitive walkthrough (CW) is a method of inspecting the usability of an interactive system that focuses on evaluating the ease of learning a new tool [30]. Its purpose is to analyze how a user thinks and behaves when they first use an interface. It is known that if users are given a choice, they prefer to learn based on exploration and observation, rather than reading manuals or following instructions [34].

The patients were allowed to interact with the Xemio app in the second part of the focus group. During this activity, the patients were given a smartphone with the app and commands about what activities to do with the app. Their impressions were collected regarding the content and usefulness of the tool. Data collection in this session was achieved using questionnaires, registering the navigation of the app, and the participant’s observations recorded by the observers.

The research team developed two questionnaires and a user test to evaluate the ease of use, effectiveness, and efficiency of the Xemio application. The session was well defined and guided by “Usability Inspection Methods, Jakob Nielsen, 1994”, taking special care with some golden rules such as one task = one action.

The order of the CW session was as follows:First questionnaire: this was aimed at understanding the degree of literacy the participants had in handling smartphone applications;User test: a selected member of the research group with expertise about the app acted as the facilitator by explaining the tasks to be completed;Second questionnaire: This was aimed at assessing the usefulness and contents of the application.

It also contrasted the answers that emerged from the first part of the focus group. A 50 min session was planned to complete this part.

### 3.3. Venue for the Discussion

The focus group was held in the living lab of the Hospital Clinic, a space dedicated to sharing experiences with patients, called Espai de Intercanvi d’Experiències (EIE) within the Hospital Clínic de Barcelona. The space for the Exchange of Experiences (EIE) is a physical space within the hospital that facilitates reflection, rethinking, and co-creating solutions to improve care services and increase their value from the patient’s perspective.

### 3.4. Data Analysis

The content analysis was carried out by coding the four thematic categories proposed, grouping and classifying the comments as positive and negative, locating the areas of interest, collecting the scores from the questionnaires, and analyzing the fluidity of navigation in the app.

## 4. Results and Reporting

The results are organized into three distinct sections. The first section includes an analysis of the results from the first session of the focus group. The second section focuses on the analysis of the questionnaires and the tasks performed with the Xemio app. Finally, the third section compares the results from both the first and second sessions.

### 4.1. The Capture of Information and Follow-Up Needs

The topic of the first part of the session was symptoms from the treatment (side effects) and their intensity. The conversation focused on treatment effects on body image, such as hair loss, spots on the skin, weight gain, and increased sweating. Afterward, the moderator directly asked about other side effects such as the effect on sleep, sexual life, or nutrition. Three participants pointed out that they experienced a metallic taste when eating food. In addition, some participants pointed out a weight loss at the beginning of the treatment that was recovered later. Topics related to surgery side effects, especially lymphedema (cork-like tenderness in the arm), were also mentioned in the discussion without going into much detail. Finally, two patients were referred for mental focus and memory problems. The focus participants maintained an objective and positive attitude throughout the discussion. 

During the session, the moderator collected most of the relevant information in a Metaplan board meeting, which constituted four main topics: symptoms, side effects, services, and news about cancer treatment advancements (Table 1).

#### 4.1.1. Textual Phases Catch from Patients

In the second part of the session, the moderator focused on how patients manage the treatments’ side effects. During that section, the participants recalled digestive side effects, nausea, mouth sores, skin burns from radiotherapy, fever, fatigue, and general malaise. However, most patients claimed to have received complete information on managing their symptoms from the hospital oncology staff. The positive perception was that they had been fully informed and had help when needed. 

In the third part of the session with the moderator, the patients of the group were asked about which services outside the hospital they used during their treatment and about their participation in activities carried out by patient associations. The first thing the participants mentioned was information of a practical nature to adapt to their new reality, such as the location of stores where they could buy wigs and scarves. The youngest patient admitted searching for terminology on the Internet. One of the older patients explained how she signed up for adult classes at the university. One of the patients expressed that she had attended a patient association session of the “Kálida” space at the Sant Pau hospital in Barcelona. When asked about the reason for not participating in patient association activities, they replied that the hours were unsuitable for them and that they maintained other personal activities.

The fourth section of the discussion with the moderator was about news consumption preferences. Participants were asked about the need for the consumption of specific news. The participants expressed that they thought there was an excess of information on the Internet. Another conversation topic about their cancer was information from conventional media that created false expectations. When asked about topics of interest to generate news, the general agreement was the preference for practical news with content such as nutrition and aesthetics tips and an agenda for group activities.

#### 4.1.2. Cognitive Walkthrough Test with Users

After a short break, the second session of the focus group was presented to the patients. This session started with a pre-test to find out the participants’ everyday use of Information Communication Technologies (ICT) resources. The results of this questionnaire are shown in Table 2.

A digital generation gap is visible in the use of tools by age, with the patients of the age group of 50 years being the most likely to use Internet tools and the older patients being less likely to use Internet tools.

### 4.2. Results of the Xemio App User Test

Observational comments on required tasks:Task (1) Find a side effect in Xemio: Patients were asked to find and record side effects in the app. P1, P3, and P4 had no difficulty, P2 had many navigation issues, and P5 also had some difficulties. They generally believed that navigating the side effects area and move intensity and recommendations could be more intuitive. Patients were looking for specific effects that were not included in the app (i.e. heart side effect) and expected to find a free text field where they could record these side effects; this is a use case we hadn’t developed yet.Task (2) Register a treatment in Xemio: Patients were asked to register the Intensity of effects on nails. Participant P5 could not find the option to get to the functionality to select and register an intensity.. General difficulty registering the intensity (it is not intuitive). Participants have problems returning to the previous screen when recording side effects and intensities. The image of the body to record dry skin is very well understood and the body part can be chosen; however, participants have problems understanding how to record the intensity. For example, the head only lets them select moderate intensity.Task (3) Consult information on types of cancer: There is confusion between the side menu and the bottom menu. P3 asks if she is able to zoom in. She expresses that the letters and symbols cannot be seen well.Task (4) Register for an event in the social agenda: the moderator decided not to complete this task when she realized that the patients were starting to have difficulty processing more new information and were experiencing difficulties following the pace of the session.Tasks (5) Configure my personal data and (6) Generate PDF document with my histories: it gave errors to some participants when they entered their data; however, they could access my diary.

After the experience of interacting with the Xemio app, participants were asked about their opinion of the application. 

To ensure a better representation of opinions by having more options than Yes/No or True/False answers, a seven-point Likert scale questionnaire was designed, providing participants with options to express themselves more accurately and a better representation of their assessment. The seven-point Likert Scale was also chosen to reduce the possibility of random or inconsistent responses and to avoid neutral judgments as occur with five-point Likert Scales. In general, the seven-point Likert scale can be a good choice for collecting detailed information about a participant’s evaluation. The results of this questionnaire are shown in Table 3, and the results of the open-ended questionnaire are shown in Table 4.

### 4.3. Combined Results

The results of the two parts of the study were somewhat different. In the first part, patients expressed complete confidence in the information provided by the oncology unit, describing it as accessible, complete, and understandable. In contrast, they expressed concern about information found on the Internet and the possibility of encountering false information.

In the second part of the study, after using the application, the participants viewed it as a positive addition to their existing sources of information. They expressed interest in the ease of access to information about practical events organized by other entities.

Comparing the results of each session, the participants who struggled with the tasks in the second part were the same individuals who do not use smartphones to access the Internet. Additionally, participants P2 and P5 had more difficulty navigating the application than the other participants.

## 5. Discussion

This focus group helps to choose functionalities and define the process of evolution and continuous improvement of the Xemio application. Selecting suitable candidates to participate in the focus group was essential to generate critical feedback and the necessary knowledge to identify unmet needs. A wide range of focus group participants provided valuable additional input from each participant.

The age of the patient is a key factor in determining the probability that the patient will incorporate technology, specifically this app, into their daily routine. The younger participants in the group had no difficulty navigating the app, while the older patients required assistance to complete tasks. Applications designed to support patients with cancer or chronic diseases may not be appropriate for those who have not acquired basic technology skills. As a result, these technological tools should not yet be considered a standard of care as they may exclude a significant portion of patients. 

However, mHealth applications have the potential to become a normal part of the standard of care in the near future, as more cancer patients acquire the necessary technological skills to use these tools. It is important to involve potential users from the beginning of the design process and throughout its evolution. There is currently a lack of evidence regarding patient knowledge and participation in the development and evaluation of medical applications [11,35]. Typically, technologies are presented to patients without their involvement in the design process and only later are they asked about their usefulness in clinical practice. To address this issue, it is crucial to adopt a patient-centered approach and prioritize identifying unmet needs before beginning the design process.

## 6. Principal Findings

Despite the limitations of this focus group, the results suggest that while breast cancer patients believe they receive adequate care and that the hospital services meet their needs, there is still room for improvement in patient care and support. This highlights the need for more research and efforts to enhance patient care in this field, even though patients are currently satisfied with the care they receive.

The adoption of mHealth tools, such as the Xemio app, has the potential to revolutionize the way chronic patients receive care in hospitals. By using smartphones, tablets, and wearable devices, patients can remotely monitor their health and communicate with their healthcare providers, without having to visit the hospital as frequently. This not only saves time and resources for patients but also reduces the burden on hospitals and healthcare providers, enabling them to focus on providing more complex care to those who need it most.

Additionally, mHealth tools can provide real-time health data, allowing providers to make more informed decisions about a patient’s care. This can lead to improved outcomes and a higher quality of care for patients. With the increasing availability of sophisticated mHealth technologies, the potential for improving care for chronic patients is enormous, and it is an area that is receiving increasing attention from researchers, healthcare providers, and policymakers alike. This could eventually lead to overall quality improvements in patient care. This was demonstrated during the second session of the focus group when patients expressed how much they appreciated the app and found it informative. This conversation led to the patients wishing they had the option to use the app on their phones during their initial diagnosis, treatment, and ongoing cancer process. 

### 6.1. Comparison with Prior Work

A few years ago, researchers conducted a review to evaluate the effectiveness of mHealth tools to support patients with chronic disease management [36]. The study, which referred to mHealth tools used for disease management as “mAdherence”, also explored the usability, feasibility, and acceptability of these tools. The researchers found that mAdherence tools and platforms were generally highly usable, feasible, and acceptable. However, they also pointed out that there is limited information available on how mHealth tools are designed to meet the needs of specific patient populations. For example, they noted that older patients may have difficulty traveling to a healthcare provider’s office and that mAdherence tools could ease this burden. The researchers recommended an iterative design process that includes systems and content development and multiple stages of user experience testing.

The following review article by Hamine et al. [36] found that 62 out of 107 studies explored the usability, feasibility, acceptability, or patient preferences for mAdherence interventions. The authors found that 27 studies in their search used randomized controlled trials (RCTs) to explore the impact on adherence behaviors, and significant improvements were observed in 15 of those studies. There were 16 out of 41 RCTs that showed significant differences between groups regarding effects on disease-specific clinical outcomes. The conclusion of the review article is that mHealth tools have the potential to facilitate adherence to disease management; however, the evidence to support its effectiveness is, so far, mixed.

### 6.2. Limitations

The oncologist treating the patient was present at the first session of the focus group. The presence of the oncologist may have changed what the participants revealed. They may have chosen not to share specific experiences because they thought it might affect their treatment or relationship with their doctor or the hospital.

Another limitation is that the sample size was very small and limited to a single focus group. Recent publications [37,38] suggest having at least three clusters to capture significance and saturation is necessary. Because of the COVID-19 emergency in June 2020, it was difficult to increase the number of focus group participants through recruitment and invitations to patients. The focus group was held at the Hospital Clínic de Barcelona, which was facing a shortage of resources due to the pandemic, making it challenging to schedule additional dates for the process. The focus group was carried out following all hygiene and safety regulations established by the government, and additional precautions were taken to avoid contact between participants considered to be at high risk.

## 7. Conclusions

While patients currently receive adequate care, there is always room for improvement, and mHealth tools have the potential to play a major role in enhancing patient care and support in the field of health and wellness.

Upcoming work will involve a long-term randomized pilot to investigate how using the Xemio app impacts the quality of life of breast cancer survivors, expected to be published during 2023. Further work will also involve the continuous evolution of the app to provide better and updated services to the users to support them throughout the cancer process, including patient evaluation tools, such as interviews and PREMS questionnaires.

## Figures and Tables

**Figure 1 ijerph-20-04584-f001:**
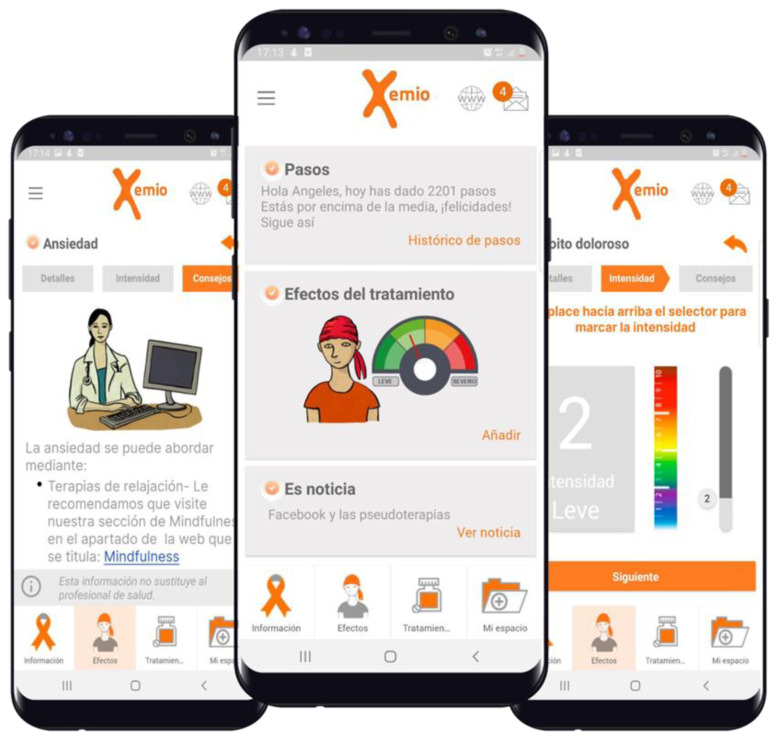
Screenshots of the mHealth Xemio App: from left to right, the screen displaying dietary hygiene advice, the main screen, and the EVA secondary effect measurement screen are shown.

**Table 1 ijerph-20-04584-t001:** Metaplan board with data collected during the session.

Topics	Anonymous Cards Written by the Participants
Treatment symptoms	difficulties sleeping
	loss of taste or appetite during treatments
	sexual problems
	reduced physical activity during treatment
	weight variations
	cramps
	hot flashes
	hair loss
	loss of memory
	ability to concentrate.
Advice on how to cope with side effects	weight change
	reliable information validated by an oncologist
	“Japanese skin” skin problems
	dry skin
	nausea
	digestion problems
	headaches
	tiredness
	fever
	general malaise
	oral pain
Services or activities needed throughout the cancer process	adequate aesthetic and beauty services
	adequate terminology
	no excessive information
	activities of patient associations
	schedules of activities
	solidarity activities
News that would be of interest during this period	avoiding fake news
	research advances
	activities
	nutrition

**Table 2 ijerph-20-04584-t002:** mHealth literacy of participants (P).

Questions Pre-Test	P1 (55 Years)	P2 (71 Years)	P3 (71 Years)	P4 (52 Years)	P5 (67 Years)
Smartphone to search for information	yes	no	no	yes	no
When a side effect occurs	email oncologist	wait until the next visit	wait until the next visit	look on the Internet wait until the next visit	wait until the next visit
Internet consultations	yes	no	yes	yes	no
Use of other services	Kalida AECC *	Adult University classes	no	no	no

* AECC: Asociación Española contra el Cáncer.

**Table 3 ijerph-20-04584-t003:** mHealth literacy of participants (P) results based on a Likert Scale questionnaire.

	Likert Scale (Num. of Responses)							
Questions	0	1	2	3	4	5	6	7
In general, do you think this app is easy to use?							2	3
Your general impression about the use of application								5
Did you know where in the app you were at all times?							1	4
Graphic design has helped you find what you were looking for?							1	4
Has the application allowed you to fulfill the tasks that were asked?							1	4

**Table 4 ijerph-20-04584-t004:** mHealth literacy of participants (P) results of open-ended questionnaires.

.	What Did You Like Most about Xemio?	What Did You Like the Least?	What Did You Think of the Graphic Design?	Do You Think You would Use Xemio?
P1	The description and information provided on symptoms were very clear. I really liked marking the intensity of the pain and the tips that I received as a result.	I like everything. There is no “minus”.	Beautiful.	Yes, in fact, I would like to read all the information about treatments, etc. Everything is very clear.
P2	Have all the information.	Small print.	Good.	Yes.
P3	Have everything on hand.	No Answer.	Very good.	Yes.
P4	Simplicity and that it is very intuitive.	How to return to the main screen.	Good.	Yes.
P5	No Answer.	No Answer.	No Answer.	No Answer.

## Data Availability

No new data were created or analyzed in this study. Data sharing is not applicable to this article.

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
