# Peer review of "mHealth Technology as a Help Tool during Breast Cancer Treatment: A Content Focus Group"

_ijerph, 2023, doi:10.3390/ijerph20054584_

Round 1

Reviewer 1 Report

The authors present the use of a mobile App as a monitoring tool for treatment side-effect and as an information resource for breast cancer patients. The authors evaluated the usefulness of this App (Xemio App) through a focus group formed by 5 breast cancer patients, by means of questionnaires, direct interaction of patients with the application and by open-ended questions. There are several issues with the present study:

- one concern lies with the methodology employed: firstly, the authors should explain why only such a small number of patients were selected for this study; this very small number of patients has an impact on the reliability of the collected data, taking also into consideration the bias mentioned by the authors;

- secondly no data on patient disease stage and type of treatment is mentioned - patients with advanced stage of disease may experience more severe symptoms following oncologic therapy compared to early stage disease or patients with less aggressive treatments;

- in section 6, page 8, lines 335-336, the authors state that "this focus group showed that care and support to patients could be significantly improved" - how was this significance demonstrated, as there is no statistical analysis performed in the study?

- in my opinion Figure 2 offers no additional information, adding no value to the manuscript; it can either be removed or transformed into a table or figure that can be easily read;

- in the Methods section, subsection 3.2 should represent the first subsection as it represents first step in the study before applying questionnaires and collecting patient-related data; furthermore subsections 3.1.1. and 3.4 seem to talk about the same think, they should be restructured into just one subsection in order to reduce redundancy;

- the authors should explain more clearly how this App will be integrated with healthcare services (for e.g.: will the attending physicians be able to see or interact with data recorded by patients)

Other minor comments:

- page 3, line 120: the numbering of subsection 3.1.1. "Cognitive walkthrough test with users (Session II)" should be 3.1.2.;

- page 1, line 32: the authors mention "during the 2000s to 1% per year", this should be rephrased for more clarity, so that readers can easily understand what this 1% represents;

- page 3, line 106: in the 3.1.1. subtitle the authors mention "...and follow-up needs Patient needs about information", this is probably a typographical error, please correct;

- English proofing of the whole text; several grammatical errors are made throughout the text;

Author Response

Dear,

Thank you very much for your review comments.

They have been helpful to us.

Reviewer 2 Report

This manuscript reports mhealth technology for breast cancer treatment. The content make sense and the idea is impressive. I recomment to accept after the following issues are solved.

1, There are many methods for the cancer or heath dignosis. The author needs to cite more papers to support the technology, such as Toward Healthcare Diagnoses by Machine Learning-Enabled Volatile Organic Compound Identification. ACS Nano, 2021, 15, 1, 894–903. 

2, What is the noverlty usiing this xemio app? any news on this technology?

3, Figure 2 looks bad, the authors can use computer to make it better in presenting

4, Is the participant enough for mhealth in table 1?

5, how to make the scale question in table 2? any reason?

6, 

Author Response

(The authors gave the same response as above.)
